# Newer-Generation EGFR Inhibitors in Lung Cancer: How Are They Best Used?

**DOI:** 10.3390/cancers11030366

**Published:** 2019-03-15

**Authors:** Tri Le, David E. Gerber

**Affiliations:** 1Department of Internal Medicine, University of Texas Southwestern Medical Center, Dallas, TX 75390-8852, USA; tri.le@phhs.org; 2Department of Clinical Sciences, University of Texas Southwestern Medical Center, Dallas, TX 75390-8852, USA; 3Division of Hematology-Oncology, Harold C. Simmons Comprehensive Cancer Center, University of Texas Southwestern Medical Center, Dallas, TX 75390-8852, USA

**Keywords:** epidermal growth factor receptor, lung cancer, osimertinib, resistance, personalized medicine, tyrosine kinase inhibitors

## Abstract

The FLAURA trial established osimertinib, a third-generation epidermal growth factor receptor (EGFR) tyrosine kinase inhibitor (TKI), as a viable first-line therapy in non-small cell lung cancer (NSCLC) with sensitizing *EGFR* mutations, namely exon 19 deletion and L858R. In this phase 3 randomized, controlled, double-blind trial of treatment-naïve patients with *EGFR* mutant NSCLC, osimertinib was compared to standard-of-care EGFR TKIs (i.e., erlotinib or gefinitib) in the first-line setting. Osimertinib demonstrated improvement in median progression-free survival (18.9 months vs. 10.2 months; hazard ratio 0.46; 95% CI, 0.37 to 0.57; *p* < 0.001) and a more favorable toxicity profile due to its lower affinity for wild-type EGFR. Furthermore, similar to later-generation anaplastic lymphoma kinase (ALK) inhibitors, osimertinib has improved efficacy against brain metastases. Despite this impressive effect, the optimal sequencing of osimertinib, whether in the first line or as subsequent therapy after the failure of earlier-generation EGFR TKIs, is not clear. Because up-front use of later-generation TKIs may result in the inability to use earlier-generation TKIs, this treatment paradigm must be evaluated carefully. For *EGFR* mutant NSCLC, considerations include the incidence of T790M resistance mutations, quality of life, whether there is a potential role for earlier-generation TKIs after osimertinib failure, and overall survival. This review explores these issues for EGFR inhibitors and other molecularly targeted therapies.

## 1. Introduction

The development of molecularly targeted cancer therapies has followed a characteristic progression in which newer agents have been developed to overcome resistance mechanisms in order to prolong disease control. In chronic myelogenous leukemia, the BCR-ABL tyrosine kinase inhibitor imatinib is a first-generation drug that has provided astounding initial response rates, good progression-free survival (PFS), and has ultimately improved long-term survival outcomes in patients [1,2,3]. However, 25–30% of patients discontinued imatinib within five years, primarily due to the development of resistance [4,5]. Thus, second-generation tyrosine kinase inhibitors (TKI), such as nilotinib and crizotinib, were designed to overcome the imatinibresistant mutations [6,7,8]. Crizotinib, has anaplastic lymphoma kinase (ALK) activity and was approved for a subset of non-small cell lung cancer found to have ALK rearrangement-driven oncogenesis. Whilst crizotinib was able to substantially improve PFS when compared to combination chemotherapy, increasing resistance to crizotinib prompted the development of later-generation ALK inhibitors—including ceritinib, alectinib, brigatinib, and lorlatinib—which had stronger activity against crizotinib-resistant cancers [9,10,11].

Later-generation molecularly targeted therapies are able to overcome resistance through the use of multiple mechanisms, including improved potency, superior penetration into sanctuary sites such as the central nervous system, and increased binding to resistance mutations. However, an emerging question is whether later-generation drugs provide superior overall outcomes in the first-line setting. The ENESTnd investigators observed that in the first-line treatment of chronic myelogenous leukemia, nilotinib had higher and earlier response rates with a lower risk of progression when compared to imatinib [12,13]. The ALEX Trial Investigators demonstrated that alectinib had superior PFS when compared to crizotinib as a first-line treatment [14]. However, overall survival still remains the most important outcome for researchers, clinicians, and patients alike.

## 2. Sequencing Targeted Therapies

The up-front use of later-generation targeted therapies raises a potential concern. Presuming that earlier-generation drugs are unlikely to have efficacy after failure of clinically superior later-generation agents, this treatment strategy implies that first-generation tyrosine kinase inhibitors would never be used in such patients. As a result, these patients would lose the period of disease control associated with earlier-generation agents. For earlier-generation epidermal growth factor receptor (EGFR) TKIs—such as erlotinib, gefitinib, and afatinib—that period generally ranges from 10 to 13 months [15,16,17,18]; for the first-generation ALK TKI crizotinib, that period ranges from 8 to 11 months [19,20,21].

Given these considerations, for the up-front use of later-generation TKIs to be a viable option, either the differential clinical benefit must exceed the lost period of disease control contributed to by the initial administration of early-generation TKIs, or there must remain a role for the use of first-generation drugs after the failure of their later-generation counterparts. Without either or both of these scenarios, the initial use of later-generation targeted therapies—despite providing a greater initial progression-free survival—could theoretically be associated with inferior overall survival.

For ALK-positive non-small cell lung cancer (NSCLC), both conditions appear to occur. The efficacy and PFS associated with the use of both ceritinib and alectinib improved when used as an initial treatment compared to when used post-crizotinib failure [22,23]. More recently, the late-generation ALK inhibitors brigatinib and lorlatinib have also demonstrated prolonged disease control [24]. Additionally, in rare cases, resistance mechanisms to later-generation drugs may in fact be sensitive to first-generation drugs, thereby allowing for the use of both therapies over time. Examples of this include the secondary *ALK* L1198F mutation and *MET* amplification, both of which may respond to crizotinib [25,26,27]. Finally, later-generation ALK inhibitors offer improved central nervous system (CNS) penetration and control of brain metastases, thus potentially improving the patients’ quantity and quality of life [28].

While questions regarding treatment sequencing have been addressed for ALK inhibitors, it was only recently that these have been studied for EGFR inhibitors. The phase 3 FLAURA trial (AZD9291 Versus Gefitinib or Erlotinib in Patients With Locally Advanced or Metastatic Non-Small Cell Lung Cancer) [29,30] assessed the efficacy of the third-generation EGFR inhibitor osimertinib versus the standard-of-care earlier-generation EGFR inhibitors (erlotinib, gefitinib, afatinib) as a first-line therapy in advanced *EGFR* mutant NSCLC. The study demonstrated the superiority of osimertinib, with a median PFS of 18.9 months versus 10.2 months for the earlier-generation EGFR inhibitors (HR 0.46, 95% CI, 0.37–0.57; *p* < 0.001). 

## 3. EGFR Inhibitors

Driving the development and investigation of osimertinib is the clinical reality of *EGFR* mutant NSCLC. With radiographic response rates exceeding 75%, the efficacies of first-generation EGFR inhibitors were greater than conventional chemotherapy in *EGFR* mutant NSCLC [31]. However, with disease control generally lasting approximately one year [32], this performance falls far short of the efficacy of BCR-ABL inhibitors for chronic myeloid leukemia, which feature five-year disease-control rates exceeding 90% [1,33]. Therapeutic resistance may be biological (i.e., due to a change in the nature of the cancer cell) or pharmacological (i.e., due to an inadequate penetration of the drug to the target tumor) [34]. The dominant biological resistance mechanism is the exon 20 T790M mutation, which occurs in up to 60% of patients with acquired resistance to EGFR TKIs [32,35]. Almost all T790M mutations are in cis with activating *EGFR* mutations, regardless of whether T790M is de novo or acquired [36]. This alteration functions as a “gate keeper” mutation, in which the significantly bulkier methionine amino acid residue replaces the threonine residue [37]. As a result of this conformational change, there is enhanced ATP affinity and reduced access of first- and second-generation EGFR inhibitors to the EGFR ATP binding pocket [38,39]. Other known biological resistance mechanisms include *MET* amplification, *EGFR* amplification, *PIK3CA* amplification, *HER2* amplification, and histologic transformation to small cell lung cancer. In up to 10% of resistant cases, the precise biologic mechanism remains unknown [40]. 

Inadequate central nervous system (CNS) penetration of EGFR TKIs is a critical consideration among pharmacologic resistance mechanisms. Approximately one-fifth of patients with advanced *EGFR* mutant NSCLC who are treated with gefinitib or erlotinib progress initially in the brain [41]. Cerebral spinal fluid (CSF) concentrations of gefitinib are less than 5% of those seen in plasma [42,43]. The role of limited drug delivery as the primary reason for CNS progression is also supported by tumor molecular profiling. Tissue from emerging or progressing brain metastases in patients receiving EGFR TKI therapy rarely demonstrate T790M resistance mutations, which is consistent with a pharmacological rather than biological mechanism [44,45]. Accordingly, the improved blood–brain barrier penetration of EGFR inhibitors emerged as an important medical need for this population. 

The categorization of EGFR inhibitors reflects their pharmacologic effects (see Table 1). First-generation EGFR inhibitors, such as erlotinib and gefitinib, bind reversibly to EGFR harboring sensitizing mutations (primarily exons 19 (deletions) and 21 (L858R substitution)) and to wild-type EGFR. The latter effect results in classic toxicities that reflect the physiological distribution of the EGFR molecule in the skin and gastrointestinal mucosa: acneiform rash (more than two-thirds of patients) and diarrhea (approximately one-third of patients) [16,17]. Second-generation EGFR inhibitors (e.g., afatinib and dacomitinib) differ by binding irreversibly to EGFR (also known as HER1) and by binding to HER2. However, they achieve minimal inhibition of exon 20 T790M mutant EGFR. As a result, these drugs may provide improved outcomes compared to first-generation EGFR inhibitors, albeit at the cost of greater toxicity causing side effects including high-grade diarrhea, rash, and paronychia [18,46]. While dacomitinib resulted in an improved overall survival compared to gefitinib in advanced *EGFR* mutant NSCLC (HR 0.76; 95% CI, 0.58–0.99; *p* = 0.04), afatinib did not achieve a significant improvement in overall survival compared to gefitinib (HR 0.86; 95% CI, 0.66–1.12; *p* = 0.26) [47,48]. It is not clear whether this is a reflection of differences in sample sizes (N = 452 in the dacomitinib trial, N = 297 in the afatinib trial), differences in post-study treatment with third-generation EGFR TKIs (15% in the afatinib trial, 10% in the dacomitinb trial), other factors, or true efficacy differences. Regardless, in contrast to third-generation EGFR TKIs, both drugs feature minimal activity when given after the failure of first-generation TKIs, with response rates of approximately 5% and a median PFS of approximately 3 months [49,50].

## 4. Osimertinib

The third-generation EGFR inhibitor osimertinib has favorable pharmacological properties for both efficacy and toxicity. It has a strong affinity for sensitizing and resistance (T790M) mutations, thereby achieving efficacy as both an initial therapy and after EGFR TKI failure. Additionally, osimertinib has a minimal inhibition of wild-type EGFR, resulting in lower rates and severity of dermatological and gastrointestinal toxicities [54]. In the phase 3 AURA3 trial, osimertinib had an objective response rate of 71% and a median PFS of 10.1 months in patients who progressed on first-line EGFR TKIs [55]. Grade 3 or greater epidermal toxicities occurred in only 1% of patients. Similar to later-generation ALK inhibitors, osimertinib achieved improved CNS penetration and efficacy in preclinical models and clinical trials. Among patients with CNS progression after first-line EGFR inhibitor therapy, the median PFS was 8.5 months with osimertinib, compared to 4.2 months with platinum-based chemotherapy [55,56]. In March 2017, the Food and Drug Administration approved the use of osimertinib for advanced *EGFR* T790M mutation-positive NSCLC after failure of first- or second-generation EGFR TKIs based on phase 1 and phase 2 results under the Breakthrough Therapy Designation Program. The use of osimertinib was further supported by the emergence of blood-based circulating cell-free tumor DNA assays. Although this technology was less sensitive than tissue-based testing, it could eliminate the need for repeated invasive biopsies in many cases [57].

In the light of osimertinib’s impressive activity against the dominant biological (i.e., T790M mutation) and pharmacological (i.e., poor CNS penetration) resistance mechanisms to earlier-generation EGFR TKIs in *EGFR* mutant advanced NSCLC, there were subsequent studies that evaluated the efficacy of osimertinib as a first-line therapy. In the AURA study, two cohorts of 60 patients with previously untreated advanced *EGFR* mutant NSCLC received osimertinib as the initial therapy. The median PFS was 20.5 months across all doses, which was approximately double that observed in studies of gefitinib, erlotinib, and afatinib [15,16,17,18,58,59]. 

Similar results were observed in the phase 3, randomized FLAURA trial. The median PFS was 18.9 months in the osimertinib arm, compared to 10.2 months with the standard of care first-line EGFR inhibitors. The PFS benefit occurred in all subgroups, including those with CNS metastases. Osimertinib was also better tolerated despite its longer duration of treatment (16.2 months with osimertinib; 11.5 months with standard of care). Thirteen percent of the patients discontinued osimertinib due to adverse events, compared to 18% of those who were receiving standard treatment (*p* = 0.15). Despite these impressive results, it remains unclear whether the initial or the post-progression use of osimertinib will provide patients with a longer life. 

## 5. The Optimal Use of Osimertinib

To determine and clarify the optimal sequencing paradigm of osimertinib, one would need a trial that directly compares the overall duration of the response to the first-line use of osimertinib versus the use of an early-generation TKI followed by the second-line use of osimertinib. To illustrate this question, consider a thought experiment in which one compares (1) the overall duration of disease control provided by the use of earlier-generation EGFR TKIs followed by osimertinib; and (2) the duration of disease control provided by the first-line use of osimertinib (see Figure 1). Acknowledging the limitations of cross-trial comparison and that the duration of disease control for earlier-generation EGFR inhibitors varies by drug, trial, and population, it is not clear whether the first-line use of osimertinib will provide the greatest period of disease control. Clearly more data are needed.

Another consideration is the distribution of EGFR resistance mechanisms, which are most often but not exclusively secondary exon 20 T790M mutations, and the differential outcome to second-line osimertinib depending on the mechanism of resistance. The T790M mutation underlies acquired resistance to early-generation EGFR TKIs in approximately 50% of patients [61]. Among patients who were T790M-positive, the response rate was 71% with a median PFS of 10.1 months [55]. Among patients who were T790M-negative, the response rate was 21% with a median PFS of 2.8 months [60]. Based on these data, osimertinib is currently approved only for cases with evidence of T790M mutation after EGFR TKI therapy. This differential effect renders these considerations particularly complex. By contrast, later-generation ALK inhibitors were initially approved for all ALK-positive patients after the failure of up-front ALK inhibitor therapy, regardless of molecular phenotype. As such, the potential benefit of sequential therapy may be better framed from the lens of a population of patients in a more nuanced thought experiment. 

In Figure 2a, we take a theoretical cohort of 10 patients and apply the paradigm of superior drug first. Using the mean PFS of 18.9 months from the FLAURA trial, the cumulative PFS of all 10 patients was 189 months. In Figure 2b,c, the same theoretical cohort of 10 patients with advanced *EGFR* mutant NSCLC are treated with erlotinib in the first line as part of the paradigm of sequential therapy. Using the mean PFS of 9.7 months from the EURTAC trial, these 10 patients would gain approximately 97 months of PFS cumulatively (labeled as a red letter ‘A’ in Figure 2a) before developing acquired resistance [15]. Based on the aforementioned estimate that 50% of patients with acquired resistance harbor the secondary T790M mutation, five of those initial 10 patients would be T790M-positive and five would be T790M-negative. In Figure 2b, we simulate the scenario in which only patients with T790M-positive disease are treated with osimertinib, as was standard practice up until the release of the FLAURA data and the first-line approval of osimertinib. Using the mean PFS of 10.1 months from the AURA3 trial, those patients who were T790M-positive would gain an additional 50.5 months of cumulative PFS. This theoretical cohort of 10 patients would gain a total 147.5 months of PFS, which was 41.5 months less than that seen in the cohort treated with the paradigm of superior drug first. In Figure 2c, we simulate the scenario in which all patients who progress on erlotinib are treated with osimertinib, independent of their T790M mutation status. The T790M-negative cohort would experience a PFS benefit of 2.8 months per person (derived from the AURA trial); thus, the T790M-negative cohort would experience a cumulative PFS benefit of 14 months. Overall, the cohort shown in Figure 2c would experience a total PFS benefit of 161.5 months, which is 27.5 months less than that of the cohort in Figure 2a. In this thought experiment, the paradigm of superior drug first may be preferable over that of sequential therapy, even if the overall duration of the response with erlotinib followed by osimertinib (9.7 months + 10.1 months = 19.8 months) in an individual T790M-positive patient was longer than that seen in which osimertinib was employed in the first line (18.9 months). 

The potential for differences in quality of life further complicated considerations of treatment sequencing. Osimertinib has a more favorable toxicity profile than earlier-generation EGFR TKIs, owing in large part to its lower affinity to wild-type EGFR. If osimertinib was used in the first-line setting, the cumulative exposure to adverse effects may be reduced. Moreover, CNS progression—a potentially morbid and costly clinical event—may be reduced with the first-line use of osimertinib. 

## 6. Acquired Resistance to Osimertinib

Regardless of whether it is used as an initial or second-line therapy, osimertinib treatment eventually fails. The biology of resistance to third-generation EGFR TKIs is varied and continues to be elucidated. Resistance mechanisms are broadly characterized as EGFR pathway-dependent and -independent [62]. EGFR pathway-dependent mechanisms include *EGFR* amplification, tertiary *EGFR* mutations (e.g., C797S, L718Q, G796D, L792F/H, L798I, L692V), and intratumoral T790M mutant/T790 wild-type heterogeneity [63]. EGFR pathway-independent mechanisms include *MET* amplification, *HER2* amplification, *FGFR1* amplification, *BRAF* mutation, *KRAS* mutation, and histologic transformation to small cell lung cancer [64,65,66,67,68,69,70,71,72,73,74,75,76,77,78]. Among these events, the tertiary *EGFR* exon 20 C797S mutation appears to be the most common, occurring in approximately one-third of patients progressing on osimertinib [65]. The C797 residue is required by third-generation EGFR TKIs to covalently bind to EGFR; when cysteine is substituted for serine, this covalent bond and EGFR inhibition cannot occur [64,79]. *EGFR* C797S mutations may occur in cis or in trans with T790M mutations [65]. Emerging strategies to overcome the C797S mutation include combination treatment with first- and third-generation EGFR TKIs [64,66,80,81] and the development of fourth-generation EGFR TKIs [70]. 

## 7. Conclusions

The FLAURA trial has clearly established osimertinib as a well-tolerated and effective systemic therapy for advanced *EGFR* mutant NSCLC in the first-line setting. Despite its robust first-line efficacy, further research is needed to determine which sequence of EGFR TKIs will ultimately provide the greatest duration of disease control, the longest overall survival, and the best quality of life. Additionally, as further insights into new treatments for acquired resistance to third-generation EGFR inhibitors emerge, osimertinib’s place in therapy may continue to evolve. 

## Figures and Tables

**Figure 1 cancers-11-00366-f001:**
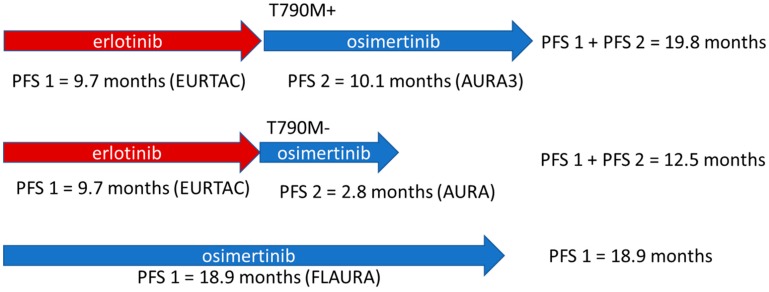
Theoretical comparison of the overall duration of response with two different sequencing paradigms. Pictured above is the sequencing strategy in which an earlier-generation EGFR TKI, erlotinib, is used in the first line, followed by the later-generation EGFR TKI osimertinib after progression in the setting of acquired T790M mutation. Progression-free survival (PFS) for first-line erlotinib is drawn from the phase 3 EURTAC trial, which compared erlotinib to conventional chemotherapy in the first-line setting [15]. The median PFS for osimertinib after progression on early-generation EGFR TKIs in T790M-positive patients was derived from the AURA3 trial [55]. The median PFS for osimertinib after progression on EGFR TKIs in T790M-negative patients was derived from the AURA trial [60]. Pictured below is the sequencing strategy in which the later-generation TKI was used in as a first-line treatment based on the PFS demonstrated in the FLAURA trial [29]. In this comparison, sequential therapy with the earlier-generation EGFR TKI followed by the later-generation TKI after progression with T790M-positivity yielded a greater overall duration of response than the use of later-generation EGFR TKI as the first-line treatment. Fully acknowledging the limitations of the cross-trial comparison, this nonetheless illustrates that the superior efficacy of later-generation therapies in the front line may not lead to superior overall survival despite a clearly superior PFS when comparing front-line therapies in isolation.

**Figure 2 cancers-11-00366-f002:**
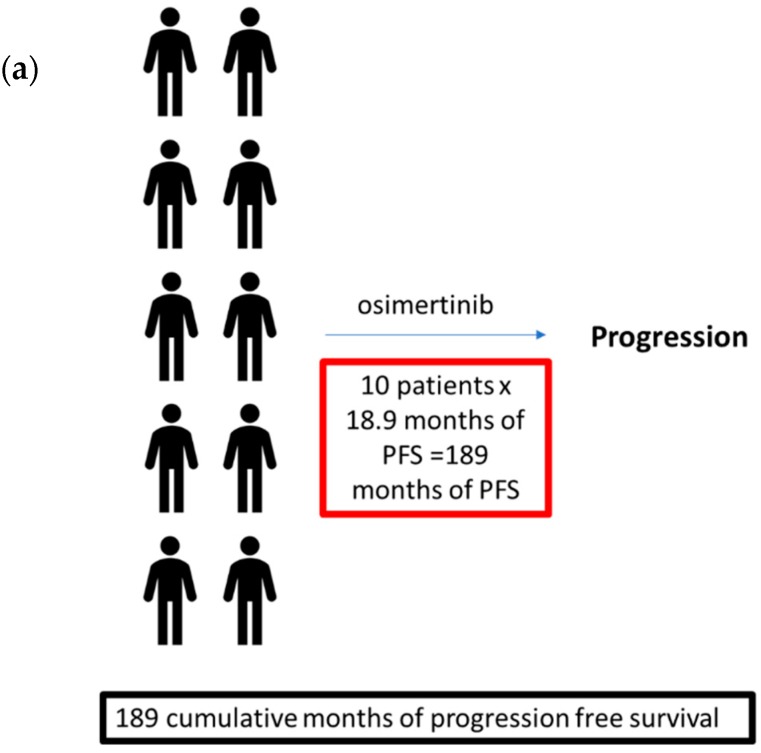
Comparison of cumulative months of progression free survival in theoretical cohorts of 10 patients treated with two sequencing strategies. (**a**) A theoretical cohort of 10 patients with advanced *EGFR* mutant NSCLC treated with osimertinib in the first line using the median PFS observed in the FLAURA trial. The estimated cumulative duration of the response for the entire cohort is calculated below. (**b**) The paradigm of sequential therapy in which a theoretical cohort of 10 patients with advanced *EGFR* mutant NSCLC treated with erlotinib in the first line followed by osimertinib if T790M-positive mutational status was present after disease progression. The median progression-free survival (PFS) for erlotinib in the first line was derived from the EURTAC trial [15]. The median PFS for osimertinib after progression on early-generation EGFR TKIs in T790M-positive patients was derived from the AURA3 trial. The median PFS for osimertinib after progression on EGFR TKIs in T790M-negative patients was derived from the AURA trial [60]. The estimated cumulative duration of response for the cohort of patients was calculated for each therapeutic step and for the entire treatment course. (**c**) A theoretical cohort of 10 patients with advanced *EGFR* mutant NSCLC treated with erlotinib in the first line followed by osimertinib regardless of their T790M mutational status after disease progression as part of the paradigm of sequential therapy. The estimated cumulative duration of response for the cohort of patients was calculated for each therapeutic step and for the entire treatment course. Acknowledging the limitation of the cross-trial comparison, this thought experiment demonstrates that the use of later-generation osimertinib may yield an overall superior duration of response in a cohort of patients even if the overall duration of response with erlotinib followed by osimertinib (9.7 months + 10.1 months = 19.8 months) in an individual patient was longer than that seen in osimertinib in the first line (18.9 months).

**Table 1 cancers-11-00366-t001:** Binding affinity, dosing, and toxicities of EGFR tyrosine kinase inhibitors.

Category	Example TKI	Wt EGFR (IC_50_ in nM)	Exon 19del (IC_50_ in nM)	L858R (IC_50_ in nM)	Exon 19del + T790M EGFRm (IC_50_ in nM)	L858R + T790M (IC_50_ in nM)	Dosing Schedule	Common Toxicities (% with Grades 3–4)
First-generation EGFR TKI	erlotinib	20 [51]	23 [52]	39 [52]	1600 [52]	>10000 [52]	150 mg once daily	diarrhea (5), rash (12), paronychia (0), stomatitis (1) [16]
Second-generation EGFR TKI	afatinib	0.5 [53]	0.2 [52]	0.2 [52]	141 [52]	196 [52]	40 mg once daily	diarrhea (14), rash (16), paronychia (11), stomatitis (8) [18]
Third-generation EGFR TKI	osimertinib	938 [52]	12 [52]	9 [52]	3 [52]	13 [52]	80 mg once daily	diarrhea (2), rash (3), paronychia (1), stomatitis (5) [29]

EGFR, epidermal growth factor receptor; TKI, tyrosine kinase inhibitor; Wt, wild-type.

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
