# Peer review of "Newer-Generation EGFR Inhibitors in Lung Cancer: How Are They Best Used?"

_cancers, 2019, doi:10.3390/cancers11030366_

Reviewer 1 Report

Newer-generation EGFR inhibitors and other targeted therapies: How are they best used? By Le and Gerber.

I very much enjoyed reading this review. It is well written and, as far as I can see, comprehensively discusses the use of EGFR inhibitors for lung cancer patients. The minor comments I have is that the title could be a bit sharper: The manuscript describes the use only in lung cancer, so perhaps include this indication in the title. ‘Other targeted therapies’ also seems a bit off; there is only the comparison between first/second and third generation inhibitors.

One or two sentences on whether resistance mutations are in cis or in trans, and how this allele specificity affects sensitivity to inhibitors would strengthen the manuscript.

Author Response

Reviewer 1:

I very much enjoyed reading this review. It is well written and, as far as I can see, comprehensively discusses the use of EGFR inhibitors for lung cancer patients. The minor comments I have is that the title could be a bit sharper: The manuscript describes the use only in lung cancer, so perhaps include this indication in the title. ‘Other targeted therapies’ also seems a bit off; there is only the comparison between first/second and third generation inhibitors.

We appreciate and agree with the Reviewer’s suggestion and have changed the title to “Newer-generation EGFR inhibitors in lung cancer:  How are they best used?” [Lines 2-3]

One or two sentences on whether resistance mutations are in cis or in trans, and how this allele specificity affects sensitivity to inhibitors would strengthen the manuscript.

This is an excellent point.  For EGFR exon 20, we have added the following statement with appropriate reference (HIkada et al, 2017):

“Almost all T790M mutations are in cis with activating EGFR mutations, regardless of whether T790M is de novo or acquired.” [Lines 93-95]

For EGFR C797S, we have added the following statement with appropriate reference (Thress et al, 2015):

“EGFR C797S mutations may occur in cis or in trans with T790M mutations.” [Line 254]

Reviewer 2 Report

Le and Gerber present a comprehensive review on EGFR-TKI with a focus on sequential use of these drugs. Overall, the review is very well written and covers the most important aspects of this topic. However, I would like to encourage the authors to acknowledge the following minor points:

- To my opinion the question of disease control in the central nervous system and the prevention or the delay in occurrence of brain metastases with newer agents, namely third generation EGFR-TKIs (as well as ALK-TKIs) might be discussed more stringently. This fact is completely missing in the abstract. For ALK positive disease the analysis from the ALEX trial by Gadgeel et al. should be cited (Gadgeel S, et al. Ann Oncol 2018)

- lines 62-67: to my opinion the most important reason to consider newer generation TKIS as a viable option for first line therapy is overall survival or their activity in difficult to treat situations (e.g. CNS disease). Although I assume that the authors would agree on that, this might be more pronounced in this review article

- for ALK positive disease and the use of alectinib in the first-line setting the J-ALEX trial as well as the ALESIA trial (ESMO 2018) might be cited. Furthermore, the randomized phase III ALTA-1L trial for brigatinib (Cambridge DR, et al. NEJM 2018) needs to be cited. 

- the second generation EGFR-TKI dacomitinib is briefly mentioned. However, discussion of the ARCHER 1050 study is missing (Mok TS, et al. JCO 2018). in this trial dacomitinib showed improved OS for patients with EGFRmut. NSCLC compared to gefitinib. The authors might briefly discuss differences between ARCHER 1050 and LUX-Lung 7 the two randomized studies comparing 2nd- vs. 1st-generation EGFR-TKIs

- two late breaking abstracts (LBA 50 and LBA 51) at the ESMO annual meeting 2018 elucidated potential resistance mechanisms in patients treated with osimertinib. I would like to encourage the authors to refer to these important studies.

Author Response

Le and Gerber present a comprehensive review on EGFR-TKI with a focus on sequential use of these drugs. Overall, the review is very well written and covers the most important aspects of this topic. However, I would like to encourage the authors to acknowledge the following minor points:

- To my opinion the question of disease control in the central nervous system and the prevention or the delay in occurrence of brain metastases with newer agents, namely third generation EGFR-TKIs (as well as ALK-TKIs) might be discussed more stringently. This fact is completely missing in the abstract. For ALK positive disease the analysis from the ALEX trial by Gadgeel et al. should be cited (Gadgeel S, et al. Ann Oncol 2018)

Excellent point.  We have added the following statement to the Abstract:

Furthermore, similar to later-generation anaplastic lymphoma kinase (ALK) inhibitors, osimertinib has improved efficacy against brain metastases.” [Lines 21-22]

This point was already mentioned in the manuscript text as follows:

Similar to later-generation ALK inhibitors, osimertinib achieves improved CNS penetration and efficacy in preclinical models and clinical trials. Among patients with CNS progression after first-line EGFR inhibitor therapy, median PFS was 8.5 months with osimertinib, compared to 4.2 months with platinum-based chemotherapy.” [Lines 135-138]

The ALEX trial by Gadgeel et al has been added as a reference to the following statement in the “Sequencing targeted therapies” section:

Due to improved efficacy when used as initial treatment compared to when used post-crizotinib failure, both ceritinib and alectinib as first-line treatment provide a progression-free survival (PFS) that appears to exceed the combined PFS of crizotinib followed by either drug.” [Lines 70-73]

- lines 62-67: to my opinion the most important reason to consider newer generation TKIS as a viable option for first line therapy is overall survival or their activity in difficult to treat situations (e.g. CNS disease). Although I assume that the authors would agree on that, this might be more pronounced in this review article

We address the activity in CNS disease in the “Sequencing targeted therapies” section:

Finally, later-generation ALK inhibitors offer improved central nervous system (CNS) penetration and control of brain metastases, characteristics that may impact both quantity and quality of life.” [Lines 76-78]

- for ALK positive disease and the use of alectinib in the first-line setting the J-ALEX trial as well as the ALESIA trial (ESMO 2018) might be cited. Furthermore, the randomized phase III ALTA-1L trial for brigatinib (Cambridge DR, et al. NEJM 2018) needs to be cited.

The J-ALEX (T Hida et al, Lancet 2017) and ALESIA (ESMO) trials are now cited in the “Sequencing targeted therapies section:

Due to improved efficacy when used as initial treatment compared to when used post-crizotinib failure, both ceritinib and alectinib as first-line treatment provide a progression-free survival (PFS) that appears to exceed the combined PFS of crizotinib followed by either drug.” [Lines 71-73]

We now mention both brigatinib and lorlatinib and promising late-generation ALK inhibitors in the “Sequencing targeted therapies” section (including referencing the ALTA-1L trial):

More recently, the late-generation ALK inhibitors brigatinib and lorlatinib have also demonstrated prolonged disease control.”

- the second generation EGFR-TKI dacomitinib is briefly mentioned. However, discussion of the ARCHER 1050 study is missing (Mok TS, et al. JCO 2018). in this trial dacomitinib showed improved OS for patients with EGFRmut. NSCLC compared to gefitinib. The authors might briefly discuss differences between ARCHER 1050 and LUX-Lung 7 the two randomized studies comparing 2nd- vs. 1st-generation EGFR-TKIs

We appreciate the Reviewer’s suggestion.  We have included these points in the “EGFR inhibitors” section (with references to the ARCHER 1050 and LUX-Lung 7 trials) as follows:

Second-generation EGFR inhibitors (e.g., afatinib and dacomitinib) differ by binding irreversibly to EGFR (also known as HER1) and by binding to HER2. However, they achieve minimal inhibition of exon 20 T790M mutant EGFR. As a result, these drugs may provide improved outcomes compared to first-generation EGFR inhibitors, albeit at the cost of greater toxicity, including high-grade diarrhea, rash, and paronychia. While dacomitinib resulted in improved overall survival compared to gefitinib in advanced EGFR mutant NSCLC (HR 0.76; 95% CI, 0.58-0.99; P=0.04), although afatinib did not achieve a significant improvement in overall survival compared to gefitinib (HR 0.86; 95% CI, 0.66-1.12; P=0.26). Whether this reflects differences in sample sizes (N=452 in the dacomitinib trial, N=297 in the afatinib trial), differences in post-study treatment with third-generation EGFR TKIs (15% in the afatinib trial, 10% in the dacomitinb trial), other factors, or true efficacy differences is not clear. Regardless, in contrast to third-generation EGFR TKIs,, both drugs feature minimal activity when given after failure of first-generation TKIs, with response rates of approximately 5 percent and median PFS approximately 3 months.”

- two late breaking abstracts (LBA 50 and LBA 51) at the ESMO annual meeting 2018 elucidated potential resistance mechanisms in patients treated with osimertinib. I would like to encourage the authors to refer to these important studies.

Both abstracts have been cited in the “Acquired resistance to osimertinib” section in conjunction with the following statements:

Whether used as initial or second-line therapy, osimertinib treatment eventually fails.  The biology of resistance to third-generation EGFR TKIs is varied and continues to be elucidated.  Resistance mechanisms are broadly characterized as EGFR pathway-dependent and –independent.  EGFR pathway-dependent mechanisms include EGFR amplification, tertiary EGFR mutations (e.g. C797S, L718Q, G796D, L792F/H, L798I, L692V), and intratumoral T790M mutant/T790 wild-type heterogeneity.”